# Multilingualism as an Object of Sociolinguistic Description

Rakesh Bhatt [1,*] and Agnes Bolonyai [2]

1   Department of Linguistics, University of Illinois, Urbana, IL 61801, USA
2   Department of English, North Carolina State University, Raleigh, NC 27695, USA
*   Correspondence: rbhatt@illinois.edu

**Abstract:** In the earlier study "Code-Switching and the Optimal Grammar of Bilingual Language Use" in 2011, we present a unified account of language use in multilingual communities using the key insight of OPTIMIZATION to capture variations between multilingual communities. This paper explores the extensions and implications of our optimality-theoretic model of multilingual grammars. We provide evidence indicating that the vast array of empirical facts of bilingual language use (code-switching) are constrained by the operation of five universal socio-cognitive constraints of multilingual grammars, and that community grammars differ from each other in terms of how they prioritize these five constraints. We provide evidence to show that the model we propose (i) accounts for bi-dialectal community grammars, as well as grammars of indigenous and transplanted multilingual communities; (ii) replicates reverse patterns of socio-grammatical differences observed earlier between indigenous and transplanted communities in terms of the relative ranking of two constraints (POWER and SOLIDARITY), linked with different indexical potentials for accruing "a profit of distinction"; and (iii) presents empirical evidence of a complete dominance hierarchy of constraint rankings, satisfying, ultimately, the desideratum of an optimality-inspired framework of assumptions, i.e., constraints are universal; constraints are in (potential) conflict with each other; constraints are violable; and the sociolinguistic grammar of bilingual language consists of the interactions between, and optimal satisfaction of, the constraints.

**Keywords:** code-switching; optimality theory; multilingual grammar; variation

## 1. Introduction

We begin this paper with full disclosure: the title of this paper is slightly plagiarized. This was the title of Charles Ferguson's plenary address at the Linguistic Summer Institute, held at the University of Illinois at Urbana-Champaign in 1978. However, the plagiarism is intentional and deliberate. We hope to show, in this essay, the intentional and deliberate aspect of this plagiarized act. Specifically, the goal of this paper is to respond to the challenges that the late Charles Ferguson described to the linguistic community in his plenary remarks in 1978.

1.   The first challenge seemed rather unremarkable: that linguists should write grammars—a rather old-fashioned thought, even for its time. The Chomskyan model of generative grammar ('standard theory') captured the imagination of every aspiring linguist willing to work with native-speakers' intuitions so as to acquire data for description and analysis. Ferguson's challenge was, however, grounded in a different paradigm of theoretical understanding of language, one where linguistic data came from community's patterns of language use. His challenge to linguists, whom he instructed to write grammars, subsumed a shift in methodological focus from the individual to the community.

2.   Ferguson's second challenge introduced a radically new idea, even for its time: that linguists should write multilingual grammars. He argued that, just as variation within a language is universal, as Labov (and many other sociolinguists) had tirelessly shown

by that time, "multilingualism in speech communities is also universal" (Ferguson 1978, p. 100). He noted that (ibid: 101):

> "... if we look around the world, including many parts of the United States, there are speech communities in which a number of different languages are part of the linguistic repertoire of the community. *Distinct languages exist side by side and are part of the whole scheme of variation of the speech community.* ... What I want to suggest ... if we are going to write a grammar of what goes on in a speech community that uses—let us say—four languages, instead of writing four separate grammars and then writing rules for when people use one language or another, *we should try to write a unified grammar in which all this variation fits somewhere.*" (*Emphasis* added)

Thus, according to Ferguson, the grammar of multilingualism is conceptually and structurally different from 'multi-monolingualisms,' a claim also echoed in François Grosjean's 1989 paper, 'Neurolinguists, beware! The bilingual is not two monolinguals in one person.'[1] Like Ferguson, Grosjean also claimed that a "bilingual is NOT the sum of two complete or incomplete monolinguals; rather, he or she has a unique and specific linguistic configuration" (Grosjean 1989, p. 3). There are strong empirical reasons witnessed across plural linguistic cultures to assume that multilinguals have a unique linguistic configuration, and that their language use offers a dynamic display of FORM and MEANING that is specific to multilingual practices.

We offer one instance, in excerpt (1) below, to show how multilinguals exploit the semantic-indexical potential of language and recruit their linguistic resources to produce different social-indexical meanings. The short excerpt below comes from a 58-year-old, upper middle-class Kashmiri woman from New Delhi, India, employing three languages in the short excerpt: *Hindi*, Kashmiri, and English.

Excerpt 1: (Bhatt and Bolonyai 2011, p. 534)

(1)    *mai jab chotii Thii* ("when I was little")
(2)    *jab meri shaadi hui* ("when I got married")
(3)    *mujhe bhii yahii lagtaa Thaa* ("I also used to think/feel")
(4)    ki myaanyan shuryan gos na kashmiri accent gasun ("that my kids should not get the 'Kashmiri accent'")
(5)    so, I spoke to them in English mainly
(6)    (pause) *bas yahii hai* ("That is it!")

What is remarkable about the data above is that, in essentially one semiotic frame, the different roles ("voice") the woman assumes are flagged by switches in languages. Very briefly, the woman in this excerpt starts speaking in *Hindi* in the role of the narrator, switches to Kashmiri in line 4, indexing the rather pervasive stance of Kashmiri diaspora mothers, and then switches to English to express the community stance that prides itself on its English language proficiency, associated with class (upper/upper-middle class) and caste (Brahmin) identities. This multilingual dynamic use of linguistic resources, expressing variable identity positionings and stance-taking decisions, is a normal, routine practice unique to multilingual contexts. Therefore, the challenge that Ferguson (1978) envisaged was for linguists to develop a grammar of multilingualism that can account for the specific configurations of FORM–MEANING pairings, as illustrated in (1) above. More generally, we must ask ourselves how we can capture this linguistic diversity in use—that is, the multilingual practice—in terms of what Ferguson called "Multilingual Grammar". Specifically, we ask: what would the architecture—the design feature—of this grammar look like?

## 2. Towards a Multilingual Grammar

In (Bhatt and Bolonyai 2011) we explored the specific configuration of multilingual grammars within an optimality-theoretic framework of assumptions, according to which the interactions between, and optimal satisfaction of, a restricted set of (ranked) socio-

cognitive constraints yield grammars of specific multilingual communities. Optimality theory has been used to address fundamental linguistic issues, such as how grammatical systems are structured and how the relationship between language universals and variability can be captured in various subfields of linguistics: phonology (McCarthy 2008), morphology (McCarthy 2006), syntax (Legendre and Vikner 2001), semantics (Hendriks and de Hoop 2001), pragmatics and natural language interpretation (Blutner and Zeevat 2004), language acquisition and learnability (Tesar and Smolensky 2000; Kager et al. 2004), historical linguistics and language change (Holt 2003), language contact phenomena (Muysken 2013), sociolinguistics (Cutillas Espinosa 2004; Kostakis 2010), and multilingualism (Bhatt and Bolonyai 2011). While the core architecture of classic optimality theory has been preserved, its extension from phonology into a range of other linguistic domains has spurred the fruitful exploration and exciting innovation of OT-based approaches. A discussion of these developments is well beyond the scope and purpose of this paper, but as an example taken from the areas closest to our current empirical focus, the incorporation of external social factors into the OT-modeling of language change and sociolinguistic variation (Kostakis 2010; Cutillas Espinosa 2004) and the centering of speaker optimization strategies as a basis for explaining the different outcomes of bilingual language contact (Muysken 2013) are worth noting.

We applied OT in our work (Bhatt and Bolonyai 2011), seeking answers to fundamental questions that are central to the study of the socio-pragmatic functions of bilingual language use (code-switching).[2] More specifically, we posed the questions of why bilinguals code-switch and why different bilingual communities reveal different patterns of code-switching (Bhatt and Bolonyai 2011, p. 522). Our purpose was to think comprehensively and theoretically about what we know of the sociolinguistic functions of code-switching and to propose that this bilingual practice and its community-specific patterning can be analyzed and accounted for in terms of a sociolinguistic grammar—that is, a system of five universal, hierarchically ordered socio-cognitive principles that, through interaction and optimal satisfaction, constrain the functional use of code-switching. The five principles deal with the conceptual-ideational, relational-interpersonal, and discourse-interactional functions of meaning making (labeled as FAITH, POWER, SOLIDARITY, FACE, and PERSPECTIVE) and were established inductively as generalizations based on over 130 functions of code-switching that we gleaned from the review of 120 studies in the literature. The methodological shift from tremendous functional variance to *relative invariance* was thus necessitated by our goal to place a theoretical order on the field and achieve generalization. At the heart of our socio-cognitive model of multilingualism lies the theoretical assumption of OPTIMIZATION, an operation (of community grammars) that selects, from a set of plausible linguistic expressions, the one that is contextually the most appropriate: the optimal output (ibid, p. 524).

Within the framework of our model, therefore, it is important to understand multilingual language practices as being underpinned and generated by the structured system of a sociolinguistic grammar, which is designed to mobilize code-switching as the most efficient linguistic resource of meaning making in the context of a given bi-/multi-lingual interaction. While the five principles (constraints) constituting our model of sociolinguistic grammar of bilingual language use (code-switching) are posited to be universal, they are locally instantiated in such a way that different communities can arrive at different optimal grammars through different constraint ranking systems and computational hierarchies. The five principles, empirically motivated by (Bhatt and Bolonyai 2011, p. 526) (emphasis added), are briefly presented below in (2):

2a.  FAITH: Social actors switch to another language if it enables them to maximize *informativity* with respect to *specificity of meaning* and *economy* of expression.

2b.  POWER: Social actors switch to another language if it enables them to maximize symbolic dominance and/or social distance.

2c.  SOLIDARITY: Social actors switch to another language if it enables them to maximize social affiliation and solidarity.

2d.  FACE: Social actors switch to another language if it enables them to maximize the effective maintenance of 'face,' or the public image of self in relation to others.

2e.  PERSPECTIVE: Social actors switch to another language if it enables them to maximize perspectivity in interactions.

Following (mainly) the theoretical logic of optimality theory (Prince and Smolensky 2004; see also Archangeli 1997; McCarthy 2007), we proposed that the exact nature of the multilingual grammar of a community derives from the community-specific ranking of the five socio-pragmatic principles, defined as meta-constraints on the code choice.[3] The differences in the hierarchical ranking of universal meta-constraints are arguably abstractions of the observable patterns of code-switching in different communities. Viewed in this light, then, "while the five meta-constraints, and the wide range of socio-pragmatic functions they underlie, are universally available for actors to draw on, the ranking of principles is community specific, and it is through participation—socialization and interaction with others—in the same community of practice that individuals come to develop an awareness and a shared grammar of locally meaningful use of two or more normatively organized codes" (Bhatt and Bolonyai 2011, p. 524). Placing inter-community variation at the center of our analysis further required us to connect community-specific variability with larger societal structures, cultural models, and ideologies. As we were short of a grand, multi-level socio-cultural theory, we offered the working hypothesis that "grammars of specific bilingual communities will vary in terms of how the constraints are ranked as a function of differences in socio-cultural norms and values; history of bilingual contact; structural position of bilingual group within the larger social historical context; and collective agency in how communities organize their bilingual resources and (re)negotiate meanings of code choice in particular socio-political economies" (Bhatt and Bolonyai 2011, p. 524).

Figure 1, below, represents the conceptual architecture of the bilingual grammatical model we proposed in Bhatt and Bolonyai (2011). In this model, the lexicons of two languages—Lex(L1) and Lex(L2)—provide the inputs to the function GEN, which takes the linguistic items from the two lexica and combines them in all possible permutations to generate potential outputs (surface structures), the candidate set. These output representations are then examined by the function EVAL, constituting a set of violable, ranked constraints, which is responsible for determining the contextual appropriateness of all the output candidates. The optimal output representation in this examination is the output that has the least serious constraint violations, i.e., violations of constraints ranked lower in the hierarchy. It thus becomes possible to capture, sociolinguistically, significant generalizations of bilingual language use, with the introduction of the idea that a sociolinguistic grammar is a set of ranked, violable constraints.

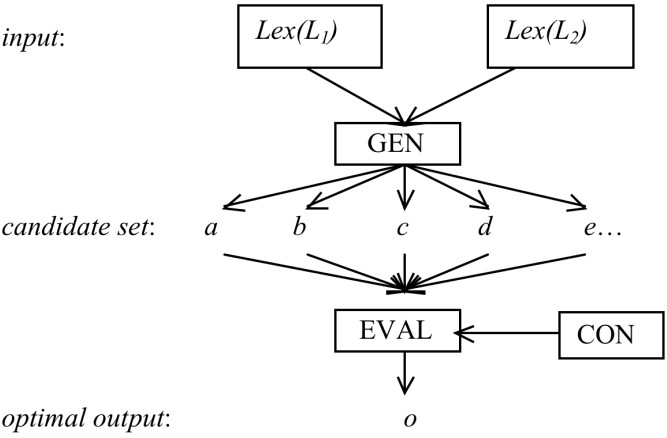

**Figure 1.** A conceptual model of bilingual grammar.[4]

In this model, then, the function EVAL is responsible for discriminating the optimal candidate from the sub-optimal ones, and it does so algorithmically. The precise mechanism

used for determining the optimality uses the following optimality-theoretic logic: Consider two grammars, Grammar A and Grammar B, and assume that both have three universal constraints {x, y, z}. Assume that Grammar A ranks these constraints in such a way that {x} dominates {y} and {y} dominates {z}. Such a grammar, clearly, imposes a total order on the constraints: x >> y >> z. Now, assume that, for a certain input, we get two competing output candidates: cand 1 and cand 2. Figure 2 shows the two competing candidates. The evaluation of the two candidates proceeds from left to right in Figure, evaluating the candidates first for the most dominant constraint and then down to the least dominant. Cand 1 violates the highest-ranking constraint {x}, indicated by "*!", which is lethal, i.e., it makes that particular candidate output contextually the least preferred. Grammar A, therefore, chooses cand 2 straightforwardly as the optimal, contextually appropriate option, indicated by "☞". Cand 1, in this evaluative matrix, is understood as "sub-optimal," or the less preferred.

Output = *cand₂* :

| Candidates | | x | y | z |
|---|---|---|---|---|
| a. | *cand₁* | *! | | * |
| ☞ b. | *cand₂* | | * | * |

**Figure 2.** Grammar A.

Using the same optimality logic, we turn to the other grammar, Grammar B. We assume that Grammar B has the same three universal constraints {x, y, z} as Grammar A above. Note that the input for Grammar B is the same as that for Grammar A, and that both Grammar B and Grammar A generate the same two competing candidate outputs: cand 1 and cand 2. What differentiates Grammar B from Grammar A is that Grammar B imposes a slightly different hierarchical ordering, viz., that {y} dominates {x} and {x} dominates {z}. The optimal output in Grammar B, as shown in Figure 3, is cand 1, because in this grammar cand 2 violates a higher-ranked constraint {y}, leading to its rejection as the optimal output.

Output = *cand₁*:

| Candidates | | y | x | z |
|---|---|---|---|---|
| ☞ a. | *cand₁* | | * | * |
| b. | *cand₂* | *! | | * |

**Figure 3.** Grammar B.

Using the logic of this grammatical architectural, Bhatt and Bolonyai (2011) offered the following optimal grammars, or constraint rankings, for the two multilingual communities:

3a.  Kashmiri multilingual community (India): Kashmiri/Hindi/English

{FAITH, PERSPECTIVE, FACE} >> POWER >> SOLIDARITY

3b.  Hungarian community (US diaspora): Hungarian/English

{FAITH, PERSPECTIVE} >> SOLIDARITY >> {FACE, POWER}

The two multilingual community grammars, Kashmiri and Hungarian, differ from each other in two specific ways. Firstly, (i) while FAITH and PERSPECTIVE are un-dominated in both of these community grammars, FACE is un-dominated in the Kashmiri community, whereas it is a constraint dominated by others in the Hungarian community. Secondly, (ii) POWER outranks SOLIDARITY in the Kashmiri community, but SOLIDARITY outranks POWER (and FACE) in the Hungarian community. In other words, the difference between Hindi–Kashmiri–English and Hungarian–English bilingual use patterns turns out to be the different ranking of SOLIDARITY vis à vis FACE and POWER. The optimality-theoretic view of multilingualism that we proposed does in fact make this very specific prediction: "different bilingual grammars will exhibit different rankings, to the extent that they differ from each other, among the proposed universal constraints" (ibid, p. 541).

### 3. Multilingual Grammars: Exemplifications and Extensions

To briefly reiterate what we were able to accomplish in Bhatt and Bolonyai (2011), we present empirical evidence from two bilingual communities in support of our claim that community bilingual grammars differ from each other in terms of how they prioritize the socio-cognitive constraints on bilingual language use. While our optimality-theoretic model of inter-community variation captured significant generalizations of bilingual language use, more empirical, cross-linguistic communal evidence must be presented, as the next methodological step, in order to support our claims about the theoretical move towards the description (and writing) of optimality-inspired grammars of bilingual use. In the remainder of the paper, we achieve precisely that: we explore several inter-related issues that arise, naturally, from our model-theoretic approach, as presented below in (4) and (5):

4. Can the model be generalized to other contexts of bilingual language use?
4i. Can it account for community grammars that are bi-dialectal, not bilingual?
4ii. Can it account for other indigenous (Kashmiri community) and transplanted (Hungarian) contexts?
4iii. Are the patterns of bilingual language use (code-switching) between indigenous and transplanted contexts replicated in the same multilingual communities?
5. Is there evidence of grammars that show a complete domination hierarchy, satisfying the desideratum of an optimality-inspired framework of assumptions?
5i. Are FAITH and PERSPECTIVE always un-dominated, and can they belong to the function GEN?

### 3.1. Inter-Dialect Switching

In terms of generalizability, as outlined in (4i) above, we first offer Cramer's (2015) study that extends our model to account for a bi-dialectal community grammar. In this paper, Cramer re-analyzes the data in Mishoe's (1995) dissertation using Bhatt and Bolonyai's (2011) optimality-theoretic framework of bilingual language use. Mishoe used the markedness model (Myers-Scotton 1993) to investigate the ways in which lower socioeconomic rural whites in a small community in the foothills of North Carolina (Cedar Falls) use dialectal-switching—specifically, between a local dialect known as Cedar Falls dialect (CFD) and a local Southern standard dialect (SSD)—in their everyday conversations with friends and family. Sifting through the various empirical details of Mishoe's bi-dialectal data, Cramer examined various possibilities of constraint interaction (and satisfaction) in order to explore the constraint rankings invested in the determination of the sociolinguistic grammar of this bidialectal community. Using pair-wise constraint rankings, she shows, systematically, how POWER outranks SOLIDARITY, and then proceeds to show, empirically, that FAITH and PERSPECTIVE outrank both. It turns out that FAITH and PERSPECTIVE are un-dominated in this community grammar. In other words, just as in the previous study of Kashmiri and Hungarian multilingual contexts (cf. Bhatt and Bolonyai 2011), the speakers in Cedar Falls utilize FAITH and PERSPECTIVE constraints to preference speaker intent and point of view. Finally, she shows that the ranking FACE is in fact the mirror image of the Hungarian–English CS situation. She concludes that, for speakers in

the Cedar Falls community, the entire range of switches between the two dialects, CFD and SSD, follows straightforwardly from a grammar that ranks the five constraints in the order given in (6), below, where FAITH and PERSPECTIVE and FACE and POWER are not ranked with respect:

6.     {FAITH, PERSPECTIVE} >> {FACE, POWER} >> SOLIDARITY

The ranking in (6) shows (i) that it is minimally different from the Kashmiri community grammar, in that the constraint FACE is not un-dominated, and (ii) that members of the Cedar Falls community prioritize FAITH and PERSPECTIVE above all other constraints. Table (7), below, represents an analysis of the data from her example (Cramer 2015, p. 181, Example 7), where a man took part in testifying in church, which required a shift from CFD to the local standard. In his testimony, as the man shares his faith journey with the congregation, he code-switches to recount the solemn moment of being saved by God and highlighting the personal significance of this religious experience. In order to show that only a shift to the local standard from CFD can accurately account for the socio-cognitive reality of code-switching in this situation, she lists four competing candidates, the outputs of GEN, which include two with switches and two mono-dialectal options.

7.     Community grammar of CFD speakers (Cramer 2015, p. 192)

| Candidates | FAITH | PERSPECTIVE | FACE | POWER | SOLIDARITY |
|---|---|---|---|---|---|
| a. ☞ Switch from CFD to local standard | | | | | * |
| b. Switch from local standard to CFD | *! | * | * | * | |
| c. Monodialectal CFD (no switch) | *! | * | | * | |
| d. Monodialectal local standard | | *! | * | | * |

*3.2. Generalizing Other Bilingual Contexts*

The optimality-theoretic approach also finds support in other contexts of bilingual language use, in both transplanted and indigenous contexts.[5] The first context we consider comes from Evensen's (2014) study of a Spanish–English bilingual community in Chicago. In this study, Evensen reviewed the past literature on code-switching among Spanish–English speakers and noticed that these studies did not consider more than one social factor in any given discourse context when, in fact, several social factors could, in principle, be implicated in the production of bilingual discourse. The goal of her paper was to discover how functional constraints play a role in shaping the grammar of Spanish–English bilingualism, specifically through the study of the interactions between the five universal meta-pragmatic constraints proposed in Bhatt and Bolonyai (2011). The hierarchy of constraints, she argued, revealed through the optimality-theoretic analysis of the code-switching of university students of Mexican heritage from Chicago, sheds light upon how bilinguals in this speech community position themselves as social actors, and which constraints allow them to assume this positioning most effectively. Using the methodological and analytical logic of optimality theory, Evensen offered the grammar, or the constraint hierarchy, of the community of Spanish heritage speakers, as shown in (8) below. The surprising feature of this community grammar, when compared with the other transplanted community grammar of Hungarian–English speakers, is that the constraint SOLIDARITY outranks POWER, but not FACE. The two other constraints, PERSPECTIVE and FAITH, remain undominated.

8.     Grammar of the Community of Spanish Heritage Speakers[6]

{PERSPECTIVE, FAITH}>>{FACE, SOLIDARITY}>>POWER

Shivaprasad's (2015) study had the specific aim of exploring and comparing the patterns in code-switching between Kannada and English among English-speaking Kannadigas (native speakers of Kannada) in Bangalore, Karnataka, India. Her study essentially replicated the results of Kashmiri–Hindi–English grammar, discussed in Bhatt and Bolonyai (2011), as presented in (9) below.

9.　　Grammar of the Kannada–English community

{FAITH, PERSPECTIVE, FACE} >> POWER >> SOLIDARITY

Combining all the results of the studies of multilingual grammars of indigenous and transplanted communities, we can observe a sociolinguistically significant generalization that is worth pointing out at this time and has to do with the relative ranking of two constraints, POWER and SOLIDARITY. In transplanted contexts, SOLIDARITY outranks POWER, while in Indigenous contexts, POWER outranks SOLIDARITY. This pattern of rankings begs two questions: Firstly, (i) will this difference in rankings hold true in comparable contexts (i.e., when code-switching data are compared in the same bilingual community in the two different contexts, indigenous and transplanted)? Secondly, (ii) why, in fact, is there a difference in ranking between these two constraints in indigenous and transplanted contexts? These questions will be addressed later in Section 3.4. For now, we merely point out a rather favorable consequence of adopting the optimality-theoretic approach to bilingual grammar, which is that it provides a hypothesis regarding how we can explain the change in multilingual practices between indigenous and transplanted communities. According to this hypothesis, the change is triggered mainly by the change in the hierarchical relations between two constraints of the sociolinguistic grammar of the multilingual community, POWER and SOLIDARITY.

Transplanted contexts: Hungarian–English, Spanish–English

{FAITH, PERSP} >> {FACE, *SOLIDARITY*} *>> POWER*

Indigenous contexts: Kashmiri–Hindi–English, Kannada–English

{FAITH, PERSP, FACE} >> *POWER >> SOLIDARITY*

The other aspect of the bilingual grammars that we can observe, thus far, has to do with the observation that FAITH and PERSPECTIVE are undominated, which begs the question of whether these two constraints possibly operate at a higher level of universality, such that they can be presumed to be un-dominated in all bilingual contexts—a rather undesirable consequence, given the architectural premise of the theory that the constraints, while universal, are violable, and even defeasible in appropriate contexts. We address this question next.

*3.3. Towards a Completely Dominant Hierarchy*

As discussed above, part of the theoretical rigor of optimality theory derives from its architectural premise, namely, that while constraints are indeed universal, they are (i) in potential conflict with each other and (ii) are NOT inviolable, or categorical. This premise yields expectations that a grammar will show (i) a complete dominance hierarchy of constraints, a theoretical desideratum, and (ii) that all universal constraints are violable, yielding different sociolinguistic grammars. As a corollary, then, the undominated constraints, FAITH and PERSPECTIVE, could, in principle, be assumed to not belong to the EVAL function but, in fact, be incorporated in the GEN function. This is a theoretically undesirable assumption.

It turns out that there are, in fact, empirical contexts where the two constraints, FAITH and PERSPECTIVE, are indeed violable, i.e., dominated by some other constraint. The first empirical context comes from a study of Korean heritage speakers (KHS) in the US (Lee 2015). In her dissertation, Lee used the optimality-theoretic model (Bhatt and Bolonyai 2011) to analyze 14 video recordings of 36 KHS who regularly attended a Korean church in a small college town in the American Midwest. The various interactions of these English–Korean bilingual speakers—aged between 24–32 years and balanced for gender—were transcribed and analyzed for their bilingual use. After combing her data for pair-wise comparisons of the different constraints, she demonstrated the familiar partial hierarchy in KHS, where SOLIDARITY outranks POWER (SOLIDARITY >> POWER), a rather unsurprising result,

given the other observed contexts of transplanted communities. Following the methodological logic of optimality theory, she systematically showed how the other three constraints outrank SOLIDARITY. There are indeed two outstanding results that stem from her analysis of the data and constitute a successful extension of the theoretical predictions of the model, including (i) the ranking between the two hitherto undominated constraints, FAITH and PERSPECTIVE, and (ii) a complete dominance hierarchy exhibited by the grammar of KHS community. After combing through the data that show an interaction between FAITH and PERSPECTIVE, she showed, using several examples (Lee 2015, pp. 129–35), how the KHS grammar prioritizes FAITH over PERSPECTIVE.

10. Interaction between FAITH and PERSPECTIVE in KHS (ibid, p. 135)

| Candidates | Faith | Perspective | Face | Power | Solidarity |
|---|---|---|---|---|---|
| ☞a. *"e, cal sayngkyessney. Coha,"* | | * | | | |
| b. "Uh, he looks good, I like him." | *! | | | | |

The other theoretically favorable result of her study is the complete dominance hierarchy of the grammar of KHS. She summarized the results of her analysis of the grammar of Korean–English heritage speakers as the constraint hierarchy given in (11), below. In most of the previous empirical contexts, discussed above, we observed only partial rankings in bilingual grammars.

11. Korean, English Heritage Speakers

FAITH >> PERSPECTIVE >> FACE >> SOLIDARITY >> POWER

To reiterate the results thus far, the question that the previous studies raised, as discussed in (5) above, is: is there evidence of grammars that show a complete domination hierarchy, satisfying the desideratum of an optimality-inspired framework of assumptions? The Korean–English bilingual grammar, (11) above, successfully addresses this question. Additionally, the constraint-ranking in (11) also confirms the optimality logic of bilingual grammars: constraints are universal, constraints are (potentially) conflicting, constraints are violable, and the observed forms of bilingual language use arise from the optimal satisfaction of conflicting constraints.

While the grammar in (11) supports an optimality-theoretic approach to bilingual grammar, we still need to address the issues of whether (i) FAITH is always un-dominated and, therefore, should in fact be part of the function GEN, and whether (ii) there is a consistent pattern of differences between indigenous (native) bilingual grammars (POWER >> SOLIDARITY) and the transplanted (displaced) bilingual grammars (SOLIDARITY >> POWER). These issues are addressed in the next section, following Karimzad's (2017, 2018) studies of Azeri multilingual communities in Iran (indigenous context) and in the US (diaspora context).

*3.4. Complete Dominance and Optimal Grammars across Contexts*

Karimzad (2018) provides a noteworthy empirical extension of the model by collecting data from Azeri–Farsi–English multilingual speakers in both Iran and the USA, providing a comparative-theoretic account of code-switching in Azeri–Farsi–English multilingual communities in the USA and Iran. The results of his data analysis reveal several interesting descriptive facts: (i) an overwhelming similarity between the multilingual grammars of Azeri communities in the USA and Iran, (ii) the fact that the constraint FAITH is dominated by another constraint, FACE, and (iii) the fact that the difference between the multilingual grammars of the two communities, while small, is significant, resulting from the interaction of SOLIDARTY and POWER, replicating the pattern observed in previous studies. Crucially, in the diaspora context, SOLIDARITY outranks POWER, but in the indigenous context, POWER outranks SOLIDARITY. For our purposes, the two important theoretical contributions that his study offers to the optimality-theoretic approaches to bilingual grammar are that: (a) FAITH is not always un-dominated and, therefore, not part of GEN, and (b)

that the salient difference between the grammars of the two Azeri communities has to do with the relative 'value' each community places on the two relational constraints: POWER and SOLIDARITY.

After diligently conducting a pair-wise evaluation of the five constraints in the two comparable communities, balanced for gender, age, and languages, Karimzad began by showing, systematically, how PERSPECTIVE outranks (SOLIDARITY, POWER), and how FAITH outranks PERSPECTIVE. He then presented evidence in order to claim that in the interaction between the two constraints, FACE and FAITH, in both Azeri communities, when they are in conflict with each other, FACE outranks FAITH. In the case of the diaspora (transplanted) Azeri community in the USA, Table (12), below, captures this ranking. It should be pointed out that even in the indigenous (native) Azeri community in Iran, the ranking of the two constraints is the same.

12. Interaction between FACE and FAITH (ibid, p. 151)

| Candidates | FACE | FAITH | PERSPECTIVE | SOLIDARITY | POWER |
|---|---|---|---|---|---|
| a. tæhrik konande | *! | | | * | |
| b. tæhrik elian | *! | | | | * |
| ☞ **arousing** | | * | | * | |

The specific rankings of the two Azeri community grammars are presented below, in (13) (Karimzad 2018, p. 153):

13a. Azeri–Farsi–English speakers in the US

FACE >>FAITH >> PERSPECTIVE >> **SOLIDARITY >> POWER**

13b. Azeri–Farsi–English speakers in Iran

FACE >>FAITH >> PERSPECTIVE >> **POWER >> SOLIDARITY**

What is indeed promising, from the perspective of an optimality-theoretic approach to bilingual language use, is that the variation in the sociolinguistic grammars of these two communities turns out to be, predictably, a function of how these communities rank the five socio-cognitive constraints differently, albeit minimally. In other words, the two grammars do not diverge much, since the two communities share the same sociolinguistic "etiquette" (Kasper 2008) of Azeris. Where the two communities differ, minimally, has to do with the relative ranking of POWER and SOLIDARITY: POWER outranks SOLIDARITY in the indigenous contexts, an empirical fact attested in other similar contexts of bilingual language use, discussed above, while SOLIDARITY outranks POWER in displaced, diaspora contexts, which is also attested in similar contexts of bilingual language use. We end the discussion in this section in order to explore the implications of this difference for movement and displacement.

A surprising advantage of pursuing an optimality approach to bilingual grammar is that it enables us to understand the linguistic effects of human displacement. When people move from one place to another, they change and acquire a new cultural-linguistic etiquette as they adapt to a new socio-cultural ecology. Bilingual speakers' ways of speaking indicate both the macro-social conditions and their subjective positions in their altered social world. The optimality model shows precisely what those micro-discursive changes in linguistic practices and indexicalities of their linguistic resources are. Specifically, the optimality grammar reveals the precise location of the change in multilingual behavior resulting from mobility—migration, or human displacement—in terms of the relative rankings of the functional constraints: POWER and SOLIDARITY. We argue, following Karimzad (2018), that the variation in the relative ranking of these two constraints in indigenous vs. displaced contexts "has to do with the particular practice that offers the profit of distinction" (Karimzad 2018, p. 154, cf. also Bourdieu 1991), enhancing "one's symbolic position within a field", i.e., "to be noticed, validated, respected, [and/or] admired" (Albright and Luke

2008, p. 14). The profit of distinction, following Bourdieu, can be secured when the speakers "are able to exploit the system of differences to their advantage" through the linguistic capital they possess. In indigenous contexts, Karimzad (ibid) argues, "the type of practice that secures the profit of distinction is the '*differentiation function*,' in terms of status/power," which is accomplished through switching to a non-local code (English). In the diaspora community, it is the *solidarity function*, indexing in-group identity, accomplished through switching to local languages (Hungarian, Spanish, Azeri), that offers the profit of distinction in order to gain "communal capital" (Karimzad 2018, p. 154, cf. Bourdieu 1986).

## 4. Conclusions

This paper presents a cogent response to the decades-old challenge that the venerable late Charles Ferguson (1978) placed on the linguistic community: to write unified grammars of bilingual (broadly construed) communities, where distinct languages are used side by side and are part of the whole scheme of variation in the speech community (ibid, p. 100). The bilingual grammar he had imagined, roughly sketched in his plenary presentation, would have had the explanatory power to account for the rich sociolinguistic variation in linguistic communities, specifically, in his initial attempt, the diglossic communities of the Arab world. These grammars are arguably part of the socially realistic paradigm of linguistic analysis that explains the variable linguistic choices of social actors, operating in routine interactions in their community life in terms of contextually sensitive linguistic constraints that are summoned, or called into action, by social conditions. As such, the unique design feature of these grammars, we argue, must fuse the computational, algorithmic, and systematically ordered with the social—*à la* Hymes' SPEAKING model, among others—in order to produce the alchemy of bilingual creativity.

Bhatt and Bolonyai (2011) was, we believe, the first attempt to write grammars of multilingual communities. The key insight that we explored, in writing grammars of multilingual communities, is the notion of the (constrained) OPTIMIZATION of sociolinguistic options. With the use of soft constraints (with variable values) that are penalized under certain conditions, we are able to capture variation in inter-community practices of multilingualism. This optimization approach predicts that code-switching—as a strategy used to creatively mobilize linguistic resources so as to exploit their functional-indexical potential—will turn out to be a more optimal option in most bilingual interactions, as it minimizes the cost function and maximizes the reward function (cf., conversational maxims, *à la* Grice 1975).

We argue that our optimality-inspired account is one of the plausible theoretical methodologies needed to write bilingual grammars. While the model-theoretic approach we proposed in 2011, aiming to account for inter-community variation, was successful in capturing significant generalizations of bilingual language use in two different communities, the next methodological step was to explore extensions and implications of the model for multilingual grammars. In other words, more empirical, cross-linguistic evidence had to be presented in order to support our claims about the theoretical move towards the description of optimality-inspired bilingual grammars. In this paper, we attempted to do just that: provide more evidence to support the idea that the vast array of empirical facts of bilingual language use (code-switching) are constrained by the operation of five universal socio-cognitive constraints of multilingual grammars, and that the community grammars differ from each other in terms of how they prioritize the five constraints. We also provided evidence to show how the model we proposed can be extended to account for (i) community grammars that are bi-dialectal, (ii) various indigenous and transplanted contexts, and (iii) the fact that the patterns of socio-grammatical differences between indigenous and transplanted communities are replicated, as well as the fact that the differences between the contexts of the two communities result from the interaction between SOLIDARTY and POWER, enabling us to capture the precise micro-discursive changes that are effected by mobility and displacement. We also provided evidence of the violations of both FAITH and PERSPECTIVE in different empirical contexts. Finally, we were able to provide evidence to

show a complete dominance hierarchy of constraint rankings, satisfying, ultimately, the desideratum of an optimality-inspired framework of assumptions. That is, constraints are universal, constraints are in (potential) conflict with each other, constraints are violable, and the sociolinguistic grammar of bilingual language use consists of the interactions between, and optimal satisfaction of, the constraints.

**Author Contributions:** Both authors contributed to the conceptualization, research, writing, and editing of this paper. All authors have read and agreed to the published version of the manuscript.

**Funding:** This research received no external funding.

**Institutional Review Board Statement:** Not applicable-this is an analysis of previously-published data, so no new board review was necessary.

**Informed Consent Statement:** Not applicable-this paper is a meta-analysis of previous works, so no new informed consent was necessary (all data here are published in other works).

**Data Availability Statement:** Not applicable.

**Acknowledgments:** First and foremost, we wish to thank the editors of this special volume for in-viting us to write this paper. This paper has evolved over the years, benefiting from comments, suggestions, and discussions with various colleagues interested in formalizing linguistic variation in code-switching practices in multilingual communities—our thanks to all of them! We would like to thank, especially, the two anonymous reviewers, and Ad Backus, James Yoon, Patrick Drack-ley, and members of the Language and Society Discussion group at the University of Illinois, Ur-bana-Champaign (LSD@UIUC) for their detailed input on the various iterations of this paper. We, of course, take full responsibility for any errors of omission and commission.

**Conflicts of Interest:** The authors declare no conflict of interest.

## Notes

[1] Grosjean (1989) argued specifically against the monolingual view of bilingualism, a result of the strong monolingual bias that has been prevalent in the language sciences. In his view, monolinguals have been the models of the "normal" speaker-hearer, and the methods of investigation developed to study monolingual speech and language have been used with little, if any, modification to study bilinguals" (p. 4).

[2] Bilingual language use has been recently termed as 'translanguaging' by some scholars, but we continue to use the standard term 'code-switching' as we see no theoretical purchase in the terminological switch (see Bhatt and Bolonyai 2022).

[3] Muysken (2013) applied a version of OT, very different from our proposal, as an integrative framework for explaining the multiple outcomes of language contact.

[4] Lex(L$_1$) & Lex(L$_2$) = Lexicon of a language; GEN= Generator function; a, b, c, . . . = competing input candidates; EVAL = Evaluator function; CON = set of universal constraints on code-switching (Bhatt and Bolonyai 2011, p. 537).

[5] Several relatively recent studies done on minor scale have replicated the model successfully in different contexts of bilingual use: in different genres—religious sermons in Arabic dialects (Alnafisah 2019); Arabic-English use in WhatsApp chats among bilinguals in Saudi Arabia (Alghamdi 2022); text message conversations between Chinese immigrants in the US gathered from WeChat (Han 2021); Tagalog-English code-switching in Facebook-Messenger-Mediated Discourse (Arnold 2014), bilingualism in Bollywood lyrics (Husain 2017), and English-Japanese bilingualism in the music/rap of MIYACHI (Kindley 2022).

[6] Ramos Arboli (2014), working on a different data-set, came up with the exact same ranking hierarchy for the grammar of Spanish-English bilinguals.

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
