# Peer review of "Multilingualism as an Object of Sociolinguistic Description"

_languages, doi:10.3390/languages7040277_

Round 1
Reviewer 1 Report
The is the authors’ attempt to expand the optimality-theoretic framework they have proposed in their earlier work for the analysis of bilingual grammars. The model presented in their 2011 was a worthwhile effort in that it aimed to bring some theoretical order to the sociolinguistic studies of codeswitching by presenting five meta-categories -- discussed in terms of principles/constraints— that motivate codeswitching, arguing that the relative hierarchical ranking of these constraints accounts for inter-community variation of bilingual grammars. The present article provides more data from different empirical sites to discuss the utility of this model in capturing not only bilingual but also bidialectal patterns of language use. However, I believe, the authors could attempt to further expand/refine their model by engaging more with recent sociolinguistic studies of multilingualism. Below, I outline my comments, concerns, and recommendations.
-The authors could clarify what they mean by ‘community’ here. Except for p. 5 where, in a quote, the notion of “community of practice” is mentioned, it appears that the authors are working with “speech community” definition throughout the article. Some would argue that this large-scale understanding of community does not do justice to the multiplicity of socialization trajectories of bilingual/multilinguals, esp. in contexts of mobility and migration, and the (intra-community) variation in their patterns of multilingual language use.
-Along similar lines, is optimization a dynamic process? That is, how does your model account for the dynamicity of multilingual practices, not only in terms of the shifts in social actors’ grammars after migration across national borders, but also the shifts in these patterns as they move across space and time in their daily lives. The question is, would it be possible for the same multilingual individual to resort to different rankings of the constraints in different contexts and in relation different (types of) people (e.g., conversations with friends, family, or community members in different contexts and occasions in diaspora)?
- Is the language (variety) the switch to which is an index of, for example, power a constant or it may index power in one context and solidarity in another? Also, if the salient difference between bilingual grammars across different communities in indigenous vs. diaspora contexts is about how they rank POWER and SOLIDARITY relative to one another, I wonder, then, whether it is more accurate to consider the rankings of the constraints, i.e., the multilingual grammars, a property of contexts rather than the communities.
- The authors may want to consider revising the “Input” part of their model. In the introduction to the work, they draw on Ferguson and Grosjean to argue that multilingualism is not the sum of multiple monolinguals. Yet, conceptualizing the input of the system as coming from two distinct lexicons belonging to different languages does not follow this argument, as far as I understand. It would make more sense if the authors used the notion of ‘repertoire’ in this regard (see Blommaert & Backus 2013), to show how it is a unitary repertoire of linguistic resources– regardless of how they have been categorized as distinct languages, dialects, etc.-- that provides the input for the optimization processes.
- Is optimization a conscious process, meaning that bilinguals are always consciously optimizing the output of their multilingual speech? Or is it a normative, habitual process rooted in social actors’ socialization histories to which they resort subconsciously in subsequent interactions? If so, is it possible to consciously calculate the costs and benefits of certain linguistic choices? How does the system account for these different processes?
- Does the model assume that there are always multilingual options from which bilinguals select the optimal output? What if, given one’s socialization, conveying certain meanings are only possible in one of their ‘languages’ or ‘dialects’, and have no option but to switch? Would that be considered a ‘switch’ to satisfy FAITH even if there are no other options?
I believe engaging with spatiotemporal approaches to the study of (multilingual) discursive practices would allow the authors to add more complexity to their characterization of bilingual grammars. Especially, the authors may find the scholarship on chronotopes and scales (see Blommaert 2010, 2015; Blommaert and De Fina 2017; Karimzad 2020, 2021; Djuraeva & Catedral 2020) useful in responding to these questions and complexifying the understandings of multilingual practices and grammars. I understand that some of this may be beyond the scope of this paper in terms of the presentation of data and analysis, but engaging with and clarifying these issues would enable the authors to expand the scopes of applicability of their model in such a way that it not only works with large-scale understandings of language, community, and mobility, but can also account for their variation across multiple small-scaled contexts and domains of everyday life. This way, the revised model would, on the one hand, open up possibilities for future sociolinguistic research, and on the other hand, provide computational linguists who have long struggled to model codeswitching with a rather straightforward framework to model bilingual practices across scales.
Author Response
Response to Reviewer 1
- ”The authors could clarify what they mean by ‘community’ here. Except for p. 5 where, in a quote, the notion of “community of practice” is mentioned, it appears that the authors are working with “speech community” definition throughout the article. Some would argue that this large-scale understanding of community does not do justice to the multiplicity of socialization trajectories of bilingual/multilinguals, esp. in contexts of mobility and migration, and the (intra-community) variation in their patterns of multilingual language use.”
Response: We disagree with the Reviewer’s assertion that contrary to our explicitly stated conception of the bilingual communities under study as communities of practice, we are working with the large-scale notion of speech community. All the studies discussed in this paper draw on empirical data collected in small communities of practice. In fact, the small size of the discourse communities is specifically referenced in some cases (e.g., line 257; line 370).
- “Along similar lines, is optimization a dynamic process? That is, how does your model account for the dynamicity of multilingual practices, not only in terms of the shifts in social actors’ grammars after migration across national borders, but also the shifts in these patterns as they move across space and time in their daily lives. The question is, would it be possible for the same multilingual individual to resort to different rankings of the constraints in different contexts and in relation different (types of) people (e.g., conversations with friends, family, or community members in different contexts and occasions in diaspora)?”
Response: Yes, Optimization is a dynamic process – a process that takes the indexical values of linguistic resources in the community’s repertoire and organizes them, in terms of their relative salience in the community (socio-cognitive constraints), to produce and re-produce a consistent set of interactional outputs that makes (intended) meaning communicatively explicit and transparent and its construal cognitively economical. The theoretical assumption in the model is that the community (socio-cognitive) grammars that optimization yields are stable, and durable – an assumption akin to Bourdieu’s concept of habitus.
As for individual (inter-speaker AND intra-speaker) variation, yes, there is within the Optimality-framework ways to accommodate the dynamic changes – this has been discussed in Bhatt & Bolonyai (2011: 543-544). Now, at this point, whether (and how) an individual bilingual speaker’s social grammar shifts as they move across different chronotopes remains an empirical question. We do not yet have any data or evidence to support or refute this hypothesis. What is significant, and is supported by extant empirical evidence, though, is that we find a strong patterned variation (i.e., the displaced vs. indigenous distinction) across a number of small-scale communities of various linguistic backgrounds.
In sum: given that we conceptualize community as a community of practice, or discourse community, it is possible for rankings to show some form of variation across these communities of practice. Moreover, just as we take social indexicality, or processes of Optimization, to be a dynamic process, so do we expect the five sociolinguistic constraints, which draw on locally
emergent or enregistered indexical meanings, to be optimized according to the variable social dynamic of any given community of speakers and thus index relevant conditions of their production.
- “Is the language (variety) the switch to which is an index of, for example, power a constant or it may index power in one context and solidarity in another? Also, if the salient difference between bilingual grammars across different communities in indigenous vs. diaspora contexts is about how they rank POWER and SOLIDARITY relative to one another, I wonder, then, whether it is more accurate to consider the rankings of the constraints, i.e., the multilingual grammars, a property of contexts rather than the communities.“
Response: We would argue the attested patterns in which POWER and SOLIDARITY are flipped in diaspora vs. indigenous communities emerge out the interaction of several factors, including how social actors respond to their contextual conditions and macro-/micro-level social positionings (aka context), and what indexical values of their linguistic resources and semiotic-ideological processes are available for speakers for dialogic/communal meaning making (changing indexicality of linguistic resources). Locating the socio-cognitive mechanism of multilingual grammars within the bilingual community may privilege perhaps a more agentive and relational/dialogic view of meaning-making, but it also reflects that the sociolinguistic generalization regrading indigenous vs. displaced community patterns resulted from a series of empirical research studies.
- “The authors may want to consider revising the “Input” part of their model. In the introduction to the work, they draw on Ferguson and Grosjean to argue that multilingualism is not the sum of multiple monolinguals. Yet, conceptualizing the input of the system as coming from two distinct lexicons belonging to different languages does not follow this argument, as far as I understand. It would make more sense if the authors used the notion of ‘repertoire’ in this regard (see Blommaert & Backus 2013), to show how it is a unitary repertoire of linguistic resources– regardless of how they have been categorized as distinct languages, dialects, etc.-- that provides the input for the optimization processes.
Response: The “Input” that is fed in the GEN function comes from two distinct sources, we label it L1 and L2 to indicate two different, distinct linguistic sources—dialects, registers, or, indeed, (standard) languages: each an archive of the cultural-historical knowledge of the communities—in bilinguals’ verbal repertoire. We have shown elsewhere (Bhatt & Bolonyai, under review) that bilingual (multilingual) speakers have access to (and exploit) distinct linguistic archives (L1, L2, L3 …) from which they draw resources to compose their meanings: we take serious exceptions to a hypothesis predicated on “a unitary repertoire of linguistic resources” for it fails to capture several significant generalizations of bilingual language use.
- ”Is optimization a conscious process, meaning that bilinguals are always consciously optimizing the output of their multilingual speech? Or is it a normative, habitual process rooted in social actors’ socialization histories to which they resort subconsciously in subsequent interactions? If so, is it possible to consciously calculate the costs and benefits of certain linguistic choices? How does the system account for these different processes?”
Response: Optimization is a sub-conscious process that engages in contextual appraisals in determining appropriate linguistic choice(s) for social actors engaged in multilingual (or even in monolingual) interactions. Social actors are not sociolinguistic dopes, they do use their verbal repertoire maximally (predicated on principles of cognitive economy and communicative efficiency) to produce desired (interactional) outcomes (see also our response above to reviewer’s comment #2).
We are not convinced of the merits of the argument that social actors’ socialization histories determine their subsequent interactions – which forces a reading of linguistic behavior that shows no evidence of novelty, or creativity needed to negotiate (social-indexical) meaning in human linguistic interaction, or that the socialization history constrains the limits of human creative linguistic potential: poetry and polemics are two quick examples of genres of linguistic art that are known to defy past (or, habitual) constraints of textual coherence and cohesion. Having said that, we agree with the reviewer that past behaviors that are sedimented into a linguistic habitus of the individual do often reflect meaning-making choices in the present -- but the present is not necessarily trapped in/by the past: if that were the case then that would mean, unfortunately, that no agency is granted to individual acts of identity, to acts of linguistic resistance and transgression. There is just too much to lose here, in terms of sociolinguistically significant generalizations, if we follow a theoretical trope that analyses present as past, and new as old.
- Does the model assume that there are always multilingual options from which bilinguals select the optimal output?
Response: No, the model (of bilingual language use) assumes that the permutations of linguistic choices range from ø to ∞, in that the GEN function can potentially create null expressions [ø] as well as any number of possibilities that are mathematically possible from two lexical sources of data set. It is the EVAL function of the model that evaluates all possible candidate sets and chooses the optimal candidate that best satisfies the constraint hierarchy (grammar) of the bilingual community.
- What if, given one’s socialization, conveying certain meanings are only possible in one of their ‘languages’ or ‘dialects’, and have no option but to switch? Would that be considered a ‘switch’ to satisfy FAITH even if there are no other options?
Response: Not sure we understand the reviewer’s comments fully, but we’ll try to answer the question as we understand it. So, if a language X, not Y, is endowed with carrying taboo words (societal constraints) then a switch from Y to X will not be an act of FAITH but an expression of FACE—to mitigate an unintended implicature (a face-threat).
- I believe engaging with spatiotemporal approaches to the study of (multilingual) discursive practices would allow the authors to add more complexity to their characterization of bilingual grammars. Especially, the authors may find the scholarship on chronotopes and scales (see Blommaert 2010, 2015; Blommaert and De Fina 2017; Karimzad 2020, 2021; Djuraeva & Catedral 2020) useful in responding to these questions and complexifying the understandings of multilingual practices and grammars. I understand that some of this may be beyond the scope of this paper in terms of the presentation of data and analysis, but engaging with and clarifying these issues would enable the authors to expand the scopes of applicability of their model in such a way that it not only works with large-scale understandings of language, community, and mobility, but can also account for their variation across multiple small-scaled contexts and domains of everyday life. This way, the revised model would, on the one hand, open up possibilities for future sociolinguistic research, and on the other hand, provide computational linguists who have long struggled to model codeswitching with a rather straightforward framework to model bilingual practices across scales.
Response: We both appreciate, and have engaged with spatiotemporal approaches in our work, but incorporating such an approach in the current paper, is indeed beyond the scope of this paper.
Reviewer 2 Report
See attached.

Author Response
- “it might be useful to indicate where the extensions of the model are (as opposed to those that mainly support the existence of the model)”
Response:
We indicated extensions:
- under Section 3.1, lines 283-84: Cramer extending the model from bilingual grammars to bi-dialectal grammar
- under Section 3.3., line 412: Lee’s ability to establish a complete dominance hierarchy of constraint ranking
- under Section 3.4., line 452: the extension made by Karimzad is indicated
- 2. “Those who have not read then earlier paper may find the use of OT language overwhelming, and a citation of a source other than Prince and Smolensky that gives a good overview of how it has been used in other subfields might be useful.”
Response: We added a paragraph under Section 2, lines 105-122, in which we briefly introduce OT and reference the main subfields and key studies that have applied OT. We also included two references in Line 166. Accordingly, the new references have been added to REFERENCES.
- “Finally, the only other issue I see is with the connection to Ferguson. I just found myself wanting more. …
Response: We added a paragraph in the CONCLUSION section.
Please find our response below to the bulleted comments:
Lines 28-31: We moved the footnote to the beginning of the first paragraph of the Introduction
Lines 34 and 44: The two challenges are numbered and pulled out
Line 70: The intro text to Excerpt 1 is in a separate paragraph
Line 7: Length of time deleted, added “short” instead
Line 149: Added missing parenthesis before (emphasis added)
Lines 312-315: we explained ‘testifying’
Lines 16-19: We added the “profit of distinction” argument (discussed in lines 515-519) in the Abstract.